# Spontaneous Plant Diversity in Urban Contexts: A Review of Its Impact and Importance

Daniela Ilie [1] and Sina Cosmulescu [2,*]

1   Doctoral School of Plant and Animal Resources Engineering, Faculty of Horticulture, University of Craiova, 13 A. I. Cuza Street, 200585 Craiova, Romania
2   Department of Horticulture & Food Science, Faculty of Horticulture, University of Craiova, 13 A. I. Cuza Street, 200585 Craiova, Romania
*   Correspondence: sina.cosmulescu@edu.ucv.ro

**Abstract:** To promote sustainability in urban green spaces, it is necessary to know the diversity of spontaneous species in these spaces. Based on the investigation and analysis of the relevant scientific literature, the diversity of spontaneous species and their importance was contextually discussed, along with the assessment of local biodiversity impact in green spaces. Studies on green spaces, spontaneous flora, biodiversity or ecosystem services, and studies on exotic species and adaptability were summarized. Finally, the existing issues regarding biodiversity and urbanization, and the role of spontaneous plants in restoring industrial areas were discussed. Based on the research carried out, it is considered that green spaces contain unique and useful biodiversity resulting from their management. Spontaneous flora can be a generator of plants with aesthetic character, which can be grown in an ecologically sound way in private gardens and natural spaces in town and village zones, with certain remarkable farming-biological characteristics (ecological plasticity, high hardiness, etc.). Biodiversity is a relevant feature of urban landscapes, offering multiple gains, and the conservation of this biodiversity in urban green spaces is fundamental and requires an integrated approach. However, urbanisation usually has a detrimental influence on local species' diversity.

**Keywords:** alien species; biodiversity; ecosystem services; green spaces; sustainability; weed





## 1. Introduction

Biodiversity protection is the main issue of the 21st century and there is a need to build capacity to support a diversity of conservation approaches that are adapted to the changing local conditions and priorities of diverse human societies [1]. Since the emergence of biodiversity as a concept, scientists have begun to use it to describe the phenomenon they study and to advocate its protection [2]. This has proved very useful in making the link between multiple worries and problems and referring to the imperiled species, natural areas conservation, and habitats preservation using a single term [2–4]. The urban ecosystem concept plays a key role in reconnecting cities to the biosphere and reducing the ecological footprint of cities while increasing resilience, the health of its citizens, and improving quality of life [3]. Ecosystem services are an ever-evolving concept. As ecosystem services become more mainstream in urban planning, the potential for urban planning to use green infrastructure to meet climate change and urban sustainability goals increases [4]. Most ecosystem services used in cities are supplied by ecosystems situated outside the city area, so it is important that cities can contribute with some of these services for themselves. This potential contribution/providing of ecosystems services can be helpful in sustaining and protecting the health of people living in cities; thus it can also enhance the quality of life in urban spaces [3]. There is the belief that urban society is disconnected from ecosystems, however, the demand for ecosystem services within the urban environment is growing, especially in the context of the COVID-19 pandemic, which has put the quality of local environments in the spotlight and people have started to see the importance of a quality

environment in their locality. The ecosystem services framework is a formal approach to describe and classify the relationship between society and nature [5]. The recreational aspect is one of the most valued aspects of urban ecosystem services [3]. The recreational value of a green space depends on the biological and structural diversity and the infrastructure built. Recreational activities offer residents the possibility of directly interacting with nature and of directly experiencing ecosystem services, this being usually important for urban residents as interactions with nature are limited in urban environments [5]. Cultural ecosystem services include aesthetic benefits that can be linked to reducing stress and increasing physical and mental health, but also psychological services that enrich human life with meaning and emotion. The vicinity to nature and vegetation areas provides multiple opportunities for physical exercise, improved mental health, and cognitive development [2]. Cityscapes can become knowledge spheres for managing the variety of life and ecosystem benefits for all members of society, and urban green spaces can be the main representative of biodiversity and a provider of ecosystem services in urban landscapes. Interest in the value of green spaces has enjoyed a revival in urban planning and management in response to the flowering of new concepts and paradigms such as 'urban greening', 'urban ecosystem services', and 'biophilic urban design' [6]. This renewed interest in urban greening has led to a number of policy developments towards protecting and enhancing the values of green spaces, especially given that climate change will have a substantial impact on the composition of biodiversity. An impact on biodiversity is also caused by urbanisation, an almost permanent change in land use that is harming the local natural ecosystem [7] and is often regarded as an immense transformative process in society [8] and a major threat to global biodiversity [9]. Profound differences in species diversity are detectable between intra-urban localities and are confirmed by a large number of studies analysing the distribution of numerous taxonomic groups in cities globally [10–12]. Species-area relationships generally have a large impact on biodiversity and, consequently, the area is an important factor explaining intra-urban biodiversity. Vegetation also plays an important role for intra-urban biodiversity and it is necessary to analyse the responses of different groups to these habitat features determined by taxon-specific requirements. The study by Beninde et al. [13] confirm the overall positive effect of vegetation island size on biodiversity in cities, which has often been postulated based on the general validity of species–area relationships. A heterogeneous vegetation structure is ideal for promoting biodiversity in urban green spaces [13]. Urban parks represent an important part of the complex network of urban ecosystems, which provide significant ecosystem services but require permanent attention. Urban greening is requiring urban planners to look for sustainable forms of urban green spaces of strategic importance to create well-being for the city's inhabitants; the effect of such research is the idea of using spontaneous vegetation in the design of green spaces. Chand et al. [14] consider that the variety of artificially created habitat structures have provided the framework for the development of spontaneous plants, which are much more easily adaptable to climatic conditions and require little care. The literature focuses on key questions of genetic resources, their conservation in green spaces, and maintaining high levels of urban biodiversity. To promote sustainability in urban green spaces, knowledge of the importance, diversity, and impact of wild flora requires some clarification. The following is a review of the responses provided in the literature on two points: (i) the diversity of spontaneous species in green spaces and its importance, and (ii) assessing the impact of local biodiversity in green spaces.

## 2. The Diversity of Spontaneous Species in Green Spaces and Its Importance

### 2.1. Green Spaces and Spontaneous Flora

Spontaneous flora is a source of plants with decorative qualities and with some particular agrobiological characteristics (ecological plasticity, high hardiness, etc.). Spontaneous vegetation is a typical component of any urban environment, and it consists of plants not intentionally planted by humans and not belonging to the remnants of natural habitats [15]. Spontaneous plants can respond quickly to the urban environment, given their strong

vitality [16]. Particular attention is paid to these "weeds" or "wild/spontaneous plants" in urban areas, with studies focusing on species composition, diversity, and response mechanisms to urban conditions [17,18]. The advantages of using native spontaneous flora species in green spaces are numerous: low establishment and maintenance costs, long term chromaticity, high variability, wide ecological range in urban or periurban green spaces, etc. There is a great interest in the identification and conservation of ornamental plant species from the spontaneous flora, even if very little research has been carried out in this direction [19–26]. Many of these can be used in green spaces, alongside plants already cultivated for ornamental purposes. The introduction of new species of spontaneous flora in the urban landscape is an important objective in the context of sustainable development. Mandă et al. [21] analysed the behaviour of spontaneous species with ornamental potential; *Arabis procurrens, Asplenium ruta muraria, Asplenium trichomanes, Blechnum spicant, Luzula luzuloides, Polypodium vulgare*, and *Saxifraga cuneifolia* were recommended for shaded or semi-shaded cliffs; *Arabis procurrens* for borders; and *Asplenium ruta muraria, Saxifraga cuneifolia*, and *Asplenium trichomanes* for container mini-gardens. Many of the spontaneous plants have high ornamental potential, e.g., *Acanthus balcanicus, Adonis vernalis, Aster tripolium* subsp. *pannonicus, Fritillaria meleagris, Galanthus elwesii, Hesperis pycnotricha, Limonium tomentellum, Salvia sclarea*, etc. can be used for the decoration of green spaces in urban and peri-urban city areas [27]. There are multiple studies on the diversification of the assortment of decorative plants by introducing herbaceous species of spontaneous flora into culture, as well as identifying other ways of using them [19–23,28–32]. Research into the literature has shown that a major driver of landscape preference appears to be the naturalness of a landscape, which is associated with the vegetation, the type, and the amount of human-induced change in a landscape. Native, spontaneous urban vegetation has been commonly described as exhibiting resilience and the ability to adapt to anthropogenic disturbances. Serret et al. [33] and Shwartz et al. [34] consider that cities host a higher number of vascular plant species than rural areas and most plant species in cities live in diverse habitats in parks, public gardens, lawns, riverbanks, railways, etc. Omar et al. [35] reported some particularly abundant species in urban green spaces in Paris such as *Chenopodium album, Plantago major, Senecio vulgaris, Lactuca serriola, Polygonum aviculare, Matricaria recutita, Stellaria media, Sisymbrium irio, Capsella bursa-pastoris, Sonchus oleraceus, Hordeum murinum, Taraxacum campylodes, Conyza canadensis*, and *Poa annua*. On the roofs of industrial buildings in Warsaw (Poland), Fornal-Pieniak and Chylinski [36] identified the following species of spontaneous flora: *Betula pendula, Calamagrostis epigejos, Sedum acre, Solidago canadensis, Tussilago farfara, Poa pratensis, Populus alba, Sambucus nigra*, and *Taraxacum officinale*. Madre et al. [37] found spontaneous plant species on green roofs in urban areas of Northern France, such as *Taraxacum ruderalia*, common in urban areas, *Cerastium glomeratum, Poa annua, Sonchus oleraceus, Epilobium tetragonum* and *Hypochaeris radicata*, but also species from xeric habitats such as *Saxifraga tridactylites*, protected species such as *Orchis laxiflora* but also many invasive species, e.g., *Buddleja davidii*. In Vega's and Küffer's [38] study in Zurich (Switzerland), the species *Plantago lanceolata* occurred frequently in islands of vegetation of all sizes; *Trifolium pratense* often occurred in large and medium-sized areas, while more shade-tolerant relatives such as *Geum urbanum* and *Polygonum aviculare* were very common around trees. Many of the species recorded have a high aesthetic character, e.g., *Centaurea scabiosa* or *Leucanthemum vulgare*, are known to have ecological and honey-making value, e.g., *Buphthalmum salicifolium, Echium vulgare*, or *Salvia pratensis*, or for butterflies, e.g., *Lotus corniculatus* [39]. Furthermore, the cemeteries are an obligatory component of urban landscape in human settlements around the world, in the rapid development of land use in megacities. They are extremely important for plant diversity, and the old urban cemeteries serve as a refuge for rare plant taxa and endangered species. Some threatened plant species that are extinct or critically endangered in adjacent areas can be found in these habitats [40,41].

## 2.2. Green Spaces and Biodiversity

The use and conservation of spontaneous vegetation in new, highly urbanised habitats is becoming increasingly well known, particularly in spacious ruderal sites. Planchuelo et al. [42] showed a positive correlation between vegetation island size and the occurrence of endangered species. According to Vega and Küffer [38], in densifying cities, large urban wilderness areas are becoming increasingly rare and therefore we need to better understand how spontaneous flora diversity can be promoted in the urban matrix even in small-vegetated areas. Such islands of vegetation in densely populated and built-up areas will bring biodiversity to places where the majority of urban residents live, thereby extending the potential ecosystem services of these green spaces such as air filtration, temperature reduction, storm reduction, water management, aesthetics, and improved health [43–45]. This is especially true for residents of lower socioeconomic status, who generally have reduced access to green spaces with high biodiversity [46]. The study by Vega and Küffer [38] thus also indicates that even small areas of vegetation can contribute significantly to maintaining a rich wild flora in cities, if they are frequent and sufficiently close to each other. In dense city centres, there is a high potential for expanding spontaneous flora areas. It may be less important to provide a few large wildflower habitats, but more important to ensure the regular presence of small ones, such as wildflower lawns, less frequent mowing [47,48], wildflower islands in designed and maintained gardens, car parks, vegetation discs around sufficiently large trees with healthy soil that will benefit trees in a drier and warmer urban climate [49] and vacant lots [50]. In many situations, the best course of action may be to reduce mowing, which will allow natural vegetation to regenerate over time [51,52]. The current stock of species in wild urban vegetation largely reflects the accidental or intentional introduction or colonization of species with ornamental potential in cities. Recently, city administrations have started to actively promote spontaneous flora in the urban matrix with the support of the general public, thus promoting urban biodiversity [45,53]. Hwang and Yue [54] propose that urban green spaces should contain wild spontaneous flora plants, as a new approach in their design and management, thus responding to the social concerns and requirements of a city and being able to provide a strong ecological stimulus, in harmony with the characteristics of the region. Li et al. [16] provide insights into the biodiversity and distribution patterns of spontaneous vegetation in urban parks in Beijing, in terms of sustainable design and the development of low-maintenance green spaces.

## 2.3. Green Spaces and Ecosystem Services

Borysiak et al. [55] argue that urban landscape is offering multiple ecosystem services such as local climate modification, pollination, and providing a pleasant location for socializing. The diversity of plant species is increasing the ecosystem benefits, based on the assessment of spontaneous vascular flora grown in urban gardens. Green areas should be conceived as plant diversity collection for local species in urban vegetation facilities. Spontaneous vegetation is a typical component of any urban environment. Spontaneous plants can respond quickly to the urban environment, given their strong vitality. They grow in any type of urban green space, as well as in unsuitable places such as walls, roofs, and industrial areas [56,57]. For a long time, people have defined them as "weeds", plants that sprout haphazardly, without receiving any intervention, in abandoned or untended areas and are not acceptable in parks and gardens. Fortunately, increasing awareness of the overlooked environmental benefits they produce has allowed 'weeds' to be reconsidered. For biodiversity conservation in urban ecology, 'weeds', also referred to as 'wild plants' or 'spontaneous vegetation plants', have been featured in several studies by ecologists, most of which have occurred in urban environments [15,58]. In addition, a number of experiments have been conducted. In long-term contaminated sites, the effects of spontaneous plant ecosystems on the accumulation and translocation of heavy metals have been assessed [59,60]. Since the beginning of the 21st century, "weeds" have attracted the attention of landscape designers in Europe and America for their low



maintenance, ecological benefits, and self-reproducing ability, leading to their redefinition as plants or urban wild vegetation. The idea of introducing these plants as an alternative to ornamental varieties and demonstrating their value in designing a low-maintenance sustainable landscape has been emphasized by a number of researchers [20,50,61,62]. Such an understanding would somehow change traditional forms of urban planting and make many people recognize the importance of this green infrastructure, which thrives in harsh conditions and provides substantial ecological benefits. As the largest constituents of green space, urban parks play an important role in urban ecology and recreation for citizens. Given that most areas in urban parks are occupied by artificial communities that require specific maintenance practices, it would be necessary to look for landscaping solutions that incorporate innovative plant ecosystems and landscape forms that integrate with "messy" ecosystems to enhance sustainability [63]. Little is known about the mechanism for maintaining spontaneous species diversity in greenspaces, and their association with planted vegetation is rarely studied. Balaj [64] concluded that urban vegetation and green areas provide a wide range of valuable ecological functions, natural beauty, valuable genetic resources of spontaneous flora, areas of cultural-educational, scientific and recreational importance, and the emergence of the modern concept of sustainable city raises the long-standing issue of the place and role of vegetation and green space in urban and suburban areas in Kosovo.

### 2.4. Alien Species and Adaptability

Spontaneous plant assemblies in urban areas contain more alien plant species than in rural areas. Most of them are valorised in decorative horticulture, trade being the key route for the introduction of foreign plants worldwide [65–68]. Consequently, the invasive spread of alien plants in green spaces often starts in urban areas [69]. As the majority of these guest garden plants can only subsist where they are grown under sustained management techniques, others evade and set up beyond these areas with no human support [70]. A number of these domesticated garden species have evolved as challenging intruders with harmful effects on local species' diversity [71]. Across the globe, the number of newly introduced alien species is still increasing [72]. As eradicating and limiting most of these alien species is difficult and costly, preventing first of all the introduction and then naturalization and expansion of new alien species should be a priority. Many current preventive hazard evaluation and regulations on intrusive non-native organisms are aiming at rejecting the introduction of species with a major invasive capacity into a state or region [73]. Nevertheless, in the case of plants, thousands of non-native species have already exceeded the introduction level as they are now evolving in public parks, botanical gardens, arboretums, and private green areas. Botanical gardens in Europe are home to approximately 80,000 plant taxa [74], and the European Garden Flora includes 20,000 plant taxa that are regularly cultivated in European gardens [75]. Certain of these plant species are likely to be naturalised in today's environment; however, they may need more time to spread [76,77]. In the cases of different species, todays' environmental conditions in the local areas of their planting have likely hindered their acclimatization. Nevertheless, environmental parameters will change rapidly under the unfolding climate warming [69]. It is hard to define the invasive power of species, but multiple elements are linked to naturalisation and/or invasive progress. First, the propagation pressure. For garden plants, this pressure can be quantified by the number of gardens where a species has been planted and the number of individuals per garden [78–80]. Secondly, if a low number of plant species have a turn to be invasive in areas that are located far away from their original weather range because of a change in their weather space [81], their climatic adaptation is in most cases positively related to naturalization success [82]. Thirdly, the chances are that plant species will naturalize and turn invasive in a certain area when they have experienced acclimatisation or intrusion before, in other geographical areas [83]. A key principle when choosing an ornamental species is that its environmental conditions are helping it to be cultivated in a certain area.

## 3. Assessing the Impact of Local Biodiversity in Green Spaces

### 3.1. Biodiversity and Urbanization

The biodiversity of spontaneous plants is an important component in urban ecosystems and contributes to the value of public life, for example, to the aesthetic enhancement of recreational parks. However, urbanisation usually has a detrimental influence on local species diversity. Urbanisation generates enormous environmental changes [84], separates people from nature [85], causes loss of natural habitats, and steadily reduces accessible areas for many wild flora and fauna. These factors combine to produce an overall reduction in biodiversity in an urban setting [86]. Urbanisation can influence the regional flora by altering the availability or spatial arrangement of habitats, the species pool, and evolutionary selection pressures on plant populations in urban environments [87]. The introduction of alien species through human activities into urbanised areas is another well-known consequence of urbanisation and poses a serious risk to biodiversity. The effects of alien plant species introductions are expected to accelerate as a result of increasing urbanisation [88]. Alien species introductions often lead to the extirpation of native plant species [89] and the decline of native biodiversity [90]. One study showed that the high richness of alien plant species in a human-dominated habitat corresponded with low native species richness [91]. However, another study confirmed that high richness of both native and alien plants occurred simultaneously with moderate artificial disturbance [92]. This is also stated by Kowarik [93] who supports the combination of well-established strategies aimed at the conservation of (semi)natural remains and the enhancement of native species in urban regions with approaches that recognize the contribution of new ecosystems to urban assemblies and associated species. Therefore, the question of whether there are positive relationships between native and alien species richness in urban habitats, e.g., different types of urban green spaces, remains a challenge for the scientific community [94]. The greening of urban public spaces is a growing issue in France. Urban greenspace management policies encourage the use of spontaneous flora. Urban ecology has aroused interest in spontaneous flora since the 1980s because this vegetation can be more than a planning; it is a greening tool, i.e., it can be a method to increase biodiversity and beautify urban spaces [95]. Rapid urbanization can alter urban green spaces with various effects on plant diversity [96]. Developing cities should aim to protect and improve the quantity and quality of urban greenspace with a focus on enhancing their ecosystem services, and in this regard, discovering the social drivers of urban greenspace landscape patterns and assessing their potential impacts on both cultivated and wild plants is essential for optimizing the planning, design, and management of urban greenspaces [96].

### 3.2. Spontaneous Plants and Restoration of Industrial Areas

Economic factors play a major role in the restoration and design of new urban green spaces. Large green spaces are lacking in urban industrial areas. Fornal-Pieniak and Chylinski [36] consider that spontaneous vegetation should be considered and proposed by designers and planners as a "cheap" alternative in the development of new green areas in urban landscapes. The restoration of abandoned quarries and large areas of new industrial and commercial buildings, including shopping centres, using spontaneous vegetation has also been proposed as a cheap alternative to extensive technical reclamation by other authors as well, e.g., Jochimsen [97]; Novák and Konvička [98]. Many human activities reduce plant diversity and affect the natural environment. Anthropogenic environmental factors indicate clear gradients of urbanisation. These include pollution levels, temperature rise, or the urban heat island effect and soil compaction along rural to urban roads [99,100]. Urban areas are often characterised by low biodiversity, the introduction of non-native species, and the simplification of species' composition and ecosystems' structure [88,101]. Despite massive and pervasive human disturbance, urban ecosystems can provide a variety of substrates for colonisation with spontaneous vegetation [88,102]. The study by Ilunga wa Ilunga et al. [103] highlighted endemic species such as *Bulbostylis pseudoperennis*, which show potential in restoring degraded metal-rich habitats and revegetating the problem area.

The same is true for the species highlighted by Useni Sikuzani et al. [104], namely *Leucaena leucocephala*, *Imperata cylindrica*, *Panicum maximum*, *Tithonia diversifolia*, and *Hyparrhenia* spp. which are essential for agropastoral activities; *Aloe vera*, known for its medicinal properties; while *Abelmoschus esculentus*, *Ipomoea batatas*, and *Citrus* spp. are of nutritional interest. In another respect, despite their potential ecological threats, *Tithonia diversifolia*, *Mimosa* spp. and *Eichhornia crassipes* could be targeted to restore soil fertility for agricultural production. It is therefore clear that in terms of how they are managed, alien species can still prove useful in a given ecosystem. Plants are among the taxa that are able to persist in highly anthropized environments, are found even in highly constructed environments, can be very diverse, are remarkable for their ornamental characteristics, and have an important ecological role due to being at the base of food chains [105].

### 3.3. Biodiversity in Urban Green Spaces—Impact

Spontaneous plants and spontaneous plant communities can be considered ecological assets when considering their tolerance of urban environments, their potential to be highly diverse and to support unique wildlife, and their contribution to providing ecosystem functions and regulating services. Madre et al. [37] demonstrated the importance of green roofs for spontaneous urban flora, as these areas act as habitats for native plant species and pointed out that substrate depth was the main factor influencing the diversity of colonizing plants. This factor also shaped the taxonomic and functional composition of spontaneous plant ecosystems. Sociologically, this spontaneous flora has the potential to improve human health and well-being and to connect residents with nature without the cost of establishing or maintaining plants. Although wild flora provides ecological value, within the urban matrix, it will likely continue to be viewed as low quality or degraded green space and will always be a dismissal of the continued existence of these spaces in one of their most ecologically valuable forms [106]. Sikorski et al. [107] investigated the role of these spaces for biodiversity conservation in comparison with high-maintenance urban parks and argue that urban parks and urban areas with spontaneous vegetation contribute to dust removal, temperature decrease, water storage, and biodiversity conservation. Despite the lack of infrastructure, densely vegetated, spontaneous areas offer greater benefits than traditionally maintained parks. Vega and Küffer [38] argue that the conservation and promotion of biodiversity in urban spaces has become a central concern in urban greening. However, as cities continue to densify, urban green spaces are becoming smaller and more isolated. Many hope that wild flora, along with wildlife biodiversity, can be maintained through networks of small informal green spaces. As cities continue to densify and expand, existing green spaces and the vegetation that supports them are under increasing pressure and urban biodiversity is generally declining [108,109]. To combat such pressures, cities need to understand how to help promote and maintain 'wild', spontaneous, self-reproducing flora in an intensively used urban matrix [29,42]. While formal green spaces are protected by legislation and planning policies, densification could threaten informal green spaces (such as wastelands and land reserved for future buildings) and small island patches of vegetation in particular [110]. Several studies have demonstrated that spontaneous vegetation can not only grow easily in a community, but also possesses a high capacity to withstand extreme weather conditions, such as long-term exposure to the sun, excessive rainfall, harsh winds, etc. [16,56,57]. It is worth noting that spontaneous vegetation requires low maintenance, is aesthetically pleasing, has regional characteristics and fast growth [111]. For this reason, spontaneous vegetation should be a suitable choice for urban greening in the development of new urban, ecological green spaces. In addition, few studies have tested the species composition of wild flora and their characteristics in different urban habitats; we are especially lacking research on how to use spontaneous plants, which can enrich ecological design and biodiversity within cities [57,112].

## 4. Conclusions

The literature records that global biodiversity needs to be known and quantified before it can be effectively managed and protected. Monitoring urban biodiversity is increasingly important given the growing anthropogenic pressures on biodiversity in urban areas. The notion of a city's natural system's benefits plays a key function in reattaching the urban areas to the natural habitats and thus optimizing the ecological footprint of urban areas, and in the meantime enhancing the resilience and the physical wellness of its citizens, thereby improving the living conditions. Green spaces have a complex and multidimensional structure and can provide a range of values to urban communities encompassing ecological, economic, social, and planning dimensions. Wild spontaneous flora is a source of plants with decorative qualities, which can be used in an ecologically sound way in individual gardens, in public green spaces in urban and rural areas, with some special agrobiological characteristics (ecological plasticity, high hardiness, etc.). The introduction of new decorative species of wild flora in urban landscapes is an important objective in the context of sustainable development and the enhancement of a natural resource base, along with biodiversity conservation. Research in the literature has shown that a major factor in landscape preference appears to be the naturalness of a landscape, with naturalness being associated with the vegetation, type, and amount of human-induced change present in a landscape. The literature consulted shows that there are species with ornamental potential in wild flora that also have high ecological plasticity, which is very important for the management of green spaces in the context of obvious climate change. Landscape research has regularly identified a number of physical attributes that appear to be related to affective experiences, such as pleasure, attractiveness, and preference. Of the four, naturalness has been seen as a particularly strong factor in preference, and the significance of this dimension has been demonstrated across a number of regions and cultures [113–115]. The biodiversity of spontaneous plants is an important component in urban ecosystems and contributes to the value of public life, to the aesthetic enhancement of recreational parks, and to the sustainable management of green spaces in cities. It is crucial in achieving the challenging goal of urban sustainability. More studies are needed on the species composition of wild spontaneous flora and their characteristics.

**Author Contributions:** Conceptualization, S.C. and D.I.; methodology, S.C.; validation, S.C.; investigation, D.I.; data curation, D.I.; writing—original draft preparation, S.C.; writing—review and editing, S.C.; supervision, S.C. All authors have read and agreed to the published version of the manuscript.

**Funding:** This research received no external funding.

**Institutional Review Board Statement:** Not applicable.

**Data Availability Statement:** Not applicable.

**Conflicts of Interest:** The authors declare no conflict of interest.

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
