# Peer review of "Spontaneous Plant Diversity in Urban Contexts: A Review of Its Impact and Importance"

_diversity, doi:10.3390/d15020277_

Round 1
Reviewer 1 Report
Dear authors,
I think your manuscript covers a very important topic concerning spontaneous plant taxa and the biodiversity of green urban areas. In general, it was easy for me to follow your way of thinking and presenting your ideas, although I think that the abstract and the Conclusion sections must be partly revised. I also have some questions concerning some of the ecological terms that you are using. Below you may find my general comments and suggestions, while in the attached pdf you can find some more specific comments, which I hope will be useful to you.
General comments
Your keywords are very poor. I would add some extra, like for instance “ecosystem services” since you discuss them both in the Introduction section and also in paragraph 2.3. You could also consider “weed” and/or “alien species”
I think the way you start describing “spontaneous flora” in lines 94-96 is rather confusing. Maybe a definition of what spontaneous flora is is missing. See for instance Li et al. 2019 (Diversity and influencing factors on spontaneous plant distribution in Beijing Olympic Forest Park-you already have this publication in your list). I am copying from their paper: “Spontaneous vegetation is a typical component of any urban environment, and it consists of plants not intentionally planted by humans and not belonging to the remnants of natural habitats (Cervelli, Lundholm, & Du, 2013). Spontaneous plants can respond quickly to the urban environment, given their strong vitality.”
In Paragraph 2.1 first, you talk generally about green spaces and spontaneous flora, then about Romania, and then you give some examples-case studies from other European countries. I think the order is not logical. If you want to talk about what is the situation at the national level (e.g. Romania) you should give this information at the end of the paragraph. But in this case, you may have to state that this manuscript also examines the potential use of spontaneous plant species in urban green areas in Romania. Otherwise, why to talk specifically about one country? Also, you do not mention all other green spaces that can host important biodiversity: cemeteries. There are relevant case studies that you could take into account. For instance: Yilmaz, H. et al., 2018. The role of Aşiyan Cemetery (İstanbul) as a green urban space from an ecological perspective and its importance in urban plant diversity. Urban Forestry and Urban Greening , vol.33 , 92-98.
Line 312-313: I suggest rethinking this sentence. Invasive species have so important and sometimes difficult-to-predict impacts on local biodiversity that I would not promote their further use. Of course neither all ornamental nor all alien plants are invasive, but the further use of invasive species cannot be recommended. Especially in the EU, there is specific regulation concerning alien and invasive species. Please check it and revise respectively.

Reviewer 2 Report
The manuscript is interesting and well structured. It would be convenient to be more specific in the title of the work (pdf attached). I only point out a few small thypos in the text body.

Reviewer 3 Report
According to the title ("Plant Diversity in Urban Contexts: Importance and Impact. A Review"), the manuscript analyzes the issues of diversity of spontaneous species and the impact of biodiversity in urban green spaces, based on relevant investigation of bibliographic resources in the domain. Certainly, a large audience could be interested in the topic because of its importance and significance. With the current conditions of global population growth, population migration to major urban centers, climate change, global warming, the accentuated loss of biodiversity after the continuous reduction of natural habitats for plants and animals, and the negative impact of the human factor on nature, it is of great interest to examine the utility and importance of urban green spaces and their biodiversity. Accordingly, this review is both interesting and timely as an article for Diversity journal.
Some minor adjustments could be made to the manuscript to assure a better emphasis on some of the ideas and aspects presented, as well as a better relevance for the audience.
Title
The title “Plant Diversity in Urban Contexts: Importance and Impact. A Review” seems impactful and directly addresses the proposed issue. However, the term 'Impact' at the end would require additional explanations, to clearly understand what impact is being discussed. In addition, it seems more desirable or suitable to first discuss the impact of plant diversity in the current urban contexts and then its importance. Therefore, it would be recommended to revise the title, to choose a more clearly formulated option. For example, instead “Plant Diversity in Urban Contexts: Importance and Impact. A Review” one could opt for “Plant Diversity in Urban Contexts: A Review of its Impact and Importance” or something similar.
Abstract
The abstract is well designed and structured, and adequately reflects the content of the review. A revision of it is welcome to adjust some terms, phrases, or formulation (and to ensure a good fluency and sense of the text), and to avoid inadvertence or typos. For example, revise in Line L 17 the sentence "The biodiversity of outdoor improvement is a relevant trait of the city landscape, ….". The syntagma "The biodiversity of outdoor improvement" seems inadequate, because it could be ambiguous for readers, so it can be reformulated (or replace ‘improvement’/‘outdoor improvement’; i.e. urban green spaces generally assume the human intervention).
Regarding the writing, there are many words separated by hyphens, probably because the text was taken from a previous form of the template. E.g.:
‘spon-taneous’ (Line L 11)
‘det-rimental’ (L 19)
‘am-bience’ (L 21)
These inadvertences must be reviewed in the entire manuscript and corrected.
Keywords
The three notions (biodiversity; green spaces; sustainability) included here are adequate. It could also be included 2-3 that are not found in the title, close to the topic of the review. In addition, you left a parenthesis at the end by mistake, please delete it.
Ful text of the manuscript
The structure of the manuscript, as well as the chapters and subchapters, cover the proposed basic issues and the objectives of the manuscript. As suggestions, some wording or syntagms should be revised to ensure the intended meaning and correct understanding of the subject by readers.
E.g., please see the text on Lines L 87-91: "To promote sustainability in urban green spaces, knowledge of the importance, diversity and impact of wild flora requires some clarification. The following is a review of the responses provided in the literature on two points: (i) the importance and diversity of spontaneous species in green spaces, and (ii) assessing the impact of local biodiversity in green spaces". Check the first objective, (i) the importance and diversity of spontaneous species in green spaces. Shouldn't the correct meaning be the diversity of spontaneous species in urban green spaces and its importance?
Likewise, in the title of the second chapter (“2. Importance and diversity of spontaneous species in green areas”), it should not be “2. The diversity of spontaneous species in green spaces and its importance”? It is just a revision suggestion to check and provide the best and clearest possible meaning, so as not to create confusion for readers.
Please also check the punctuation marks. At the end of the first section, 'Introduction' (L 91), there is no period. Check the entire manuscript, i.e., in another place, an additional period sign appears, which must be deleted (see L 74: “… The study by Beninde et al. [12]. confirm the overall positive effect…” (delete period after citation number, and before ‘confirm’).
Citations (and references respectively) are representative of the review and the issues addressed. But even for a short review, they could be supplemented with a few more new ones, to update the bibliographic investigation. Possibly even to issues included in the manuscript, related to the current global issues that substantiate and justify the interest in the research (the recent population of 8 billion on earth; global resource impoverishment etc.).
In addition, in some cases to argue more consistently a presented problem or statement. For example, see L 116-119: “At international level there are multiple studies on the diversification of the assortment of decorative plants by introducing herbaceous species of spontaneous flora into culture, as well as identifying other ways of using them [26]”. Here, reformulate and delete ‘At international level’ (is superfluous, or redundant). You affirm “there are multiple studies”, but use only a citation. It is obvious that the sentence requires more citations, and except Maloupa et al., 2005 (justified citation, but quite old) you will surely find enough adequate and timely resources to support better your statement.
